# Quantitative determination of lurbinectedin, its unbound fraction and its metabolites in human plasma utilizing ultra-performance LC–MS/MS

**Nicholas King** [1]*, **Soledad Garcia-Martinez**[1], **Eider Alcaraz**[1], **Alba Grisalena**[1], **Rubin Lubomirov** [2], **Raquel Altares**[2], **Carlos Fernandez-Teruel**[2], **Andrés M. Francesch**[3], **Pablo M. Avilés**[3], **Salvador Fudio**[2]

**1** Dynakin S.L. Bioanalytical Laboratory, Derio, Spain, **2** PharmaMar S.A., Clinical Development, Colmenar Viejo (Madrid), Spain, **3** PharmaMar S.A., Research and Development, Colmenar Viejo (Madrid), Spain

* nking@dynakin.com

## Abstract

### Aims

Ultra-performance liquid chromatography–tandem mass spectrometry (UPLC–MS/MS) methods to quantify total lurbinectedin, its plasma protein binding to derive the unbound fraction and its main metabolites 1′,3′-dihydroxy-lurbinectedin (M4) and *N*-desmethyl-lurbinectedin (M6) in human plasma, were developed and validated.

### Materials & methods

For lurbinectedin, sample extraction was performed using supported liquid extraction. For metabolites, liquid-liquid extraction with stable isotope–labeled analogue internal standards was used. Plasma protein binding was evaluated using rapid equilibrium dialysis. *In vitro* investigations at different plasma protein concentrations were carried out to estimate dissociation rate constants to albumin and alpha-1-acid glycoprotein (AAG).

### Results

Calibration curves displayed good linearity over 0.1 to 50 ng/mL for lurbinectedin and 0.5 to 20 ng/mL for the metabolites. Methods were validated in accordance with established guidance.

The inter-day precision and accuracy ranged from 5.1% to 10.7%, and from -5% to 6% (lurbinectedin in plasma); from 3.1% to 6.6%, and from 4% to 6% (lurbinectedin in plasma: PBS); from 4.5% to 12.9%, and from 4% to 9% (M4); and from 7.5% to 10.5%, and from 6% to 12% (M6). All methods displayed good linearity ($r^2$ >0.99). Recovery was evaluated for lurbinectedin in plasma:PBS (66.4% to 86.6%), M4 (7.82% to 13.4%) and M6 (22.2% to 34.3%).

**Data Availability Statement:** All relevant data are within the paper.

**Funding:** This work was supported by PharmaMar S.A (Colmenar Viejo, Spain). Authors employed by PharmaMar were involved in all aspects of this work.

**Competing interests:** Raquel Altares, Rubin Lubomirov, Andrés M. Francesch, Pablo Manuel Avilés and Salvador Fudio are employed by and/or have ownership interests (including patents and/or shareholding) in PharmaMar (Colmenar Viejo, Spain). Carlos Fernández-Teruel was an employee of PharmaMar at the time of the study.

The method for lurbinectedin in plasma has been applied in most clinical studies, while the plasma:PBS and metabolites methods were used to evaluate the impact of special conditions on lurbinectedin PK.

Lurbinectedin plasma protein binding was 99.6% and highly affected by AAG concentration.

## Conclusions

These UPLC–MS/MS methods enable the rapid and sensitive quantification of lurbinectedin and its main metabolites in clinical samples.

## 1 Introduction

Lurbinectedin (Zepzelca$^{®}$) (Fig 1) is a selective inhibitor of oncogenic transcription that binds preferentially to guanines located in the GC-rich regulatory areas of DNA gene promoters [1,2]. The drug thus prevents binding of transcription factors to their recognition sequences, inhibiting oncogenic transcription and leading to tumor cell apoptosis [3]. By inhibiting activated transcription in tumor-associated macrophages, lurbinectedin also affects the tumor microenvironment landscape [4]. On June 15, 2020, the US Food and Drug Administration gave accelerated approval to lurbinectedin monotherapy in small-cell lung cancer that has relapsed from platinum compound–based first-line chemotherapy.

Two bioanalytical assays used to quantify lurbinectedin in plasma have been reported to date. The first one was focused on lurbinectedin quantification in non-human plasma (i.e., mouse, rat, dog, non-human primates [NHPs] and mini-pig) [5]. The second one [6] adapted the non-human method to make it suitable for human samples collected in a mass balance study aimed at elucidating the disposition, elimination and metabolic profiling of the compound [7]. However, early- and late-phase clinical trials required a faster method to accurately quantify lurbinectedin total plasma concentration in larger sample batches.

Lurbinectedin has a low hepatic extraction ratio of 0.19 [8], and is highly protein-bound [7]. A population pharmacokinetic (PK) analysis with data from 443 cancer patients showed that plasma protein binding (PPB) affected lurbinectedin total clearance; high alpha-1-acid glycoprotein (AAG) and low albumin reduced total plasma clearance by 28% and 20%, respectively [8]. Consequently, lurbinectedin unbound concentration becomes a more relevant exposure metric than total concentration when relating safety and efficacy outcomes to treatment with lurbinectedin by means of exposure-response models [9]. For this end, *in vitro* PPB experiments were conducted to estimate the dissociation rate constants for AAG and albumin, to be used along with total concentration and the patient's individual AAG and albumin levels to derive lurbinectedin unbound concentration. The PPB method for these *in vitro* experiments was rapid equilibrium dialysis (RED), in which plasma was diluted with phosphate-buffered saline (PBS). Therefore, a revalidation of the plasma method to quantify lurbinectedin in plasma:PBS was required.

*In vitro* and *in vivo* experiments describing the disposition and metabolism pathway of lurbinectedin in nonclinical species (rats and NHPs), as well as in advanced cancer patients treated with radio-labeled lurbinectedin in the mass balance study, have been described elsewhere [7]. Plasma metabolic profiling demonstrated that major circulating metabolites (% compared with parent compound) were *N*-desmethyl-lurbinectedin or metabolite 6 (M6; 0.4% and 10.4% in NHPs and patients, respectively) and 1′,3′-dihydroxy-lurbinectedin or metabolite 4 (M4; 0.9% and 14.3% in NHPs and patients, respectively). Consequently, methods for the

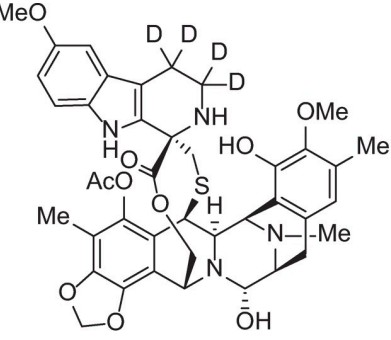

Lurbinectedin (PM01183,
$C_{41}H_{44}N_4O_{10}S$, MW: 784.87)

Lurbinectedin IS (PM040038,
$C_{41}H_{40}D_4N_4O_{10}S$, MW: 788.89)

1',3'-Dihydroxy-lurbinectedin
(M4, $C_{40}H_{44}N_4O_{10}S$, MW: 772.87)

1',3'-Dihydroxy-lurbinectedin-d4
(M4-d4, $C_{40}H_{40}D_4N_4O_{10}S$, MW: 776.87)

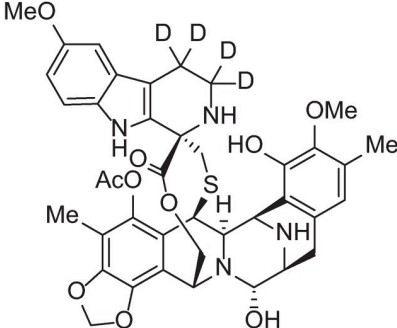

N-Desmethyl-lurbinectedin
(M6, PM030047, $C_{40}H_{42}N_4O_{10}S$, MW: 770.85)

N-Desmethyl-lurbinectedin-d4
(PM190539, $C_{40}H_{38}D_4N_4O_{10}$, MW: 774.87)

**Fig 1. The chemical structures of lurbinectedin, lurbinectedin metabolites and their stable isotope–labeled internal standards.**

quantification of metabolites M4 and M6 were developed and validated to be used in the dedicated drug-drug interaction and hepatic impairment studies, to evaluate how these patients' conditions may affect lurbinectedin metabolism.

Therefore, this paper is aimed at describing the analytical methods used to quantify total and unbound lurbinectedin in plasma and metabolites M4 and M6, as well as the *in vitro* PPB experiments to calculate the dissociation rate constants for AAG and albumin, that led to a thorough characterization of plasma PK of total and unbound lurbinectedin and its most prominent related products. These methods were successfully validated in accordance with the

US Food and Drug Administration and the European Medicines Agency guidelines on bioanalytical method validation, and can therefore be applied in pharmacological clinical studies [10,11].

## 2 Materials and methods

### 2.1 Chemicals and materials

Lurbinectedin (PM01183), 1′,3′-dihydroxy-lurbinectedin (M4), *N*-desmethyl-lurbinectedin (M6 or PM030047) and their corresponding stable isotope–labeled analogues PM040038, M4-d$_4$ and M6-d$_4$ (or PM190539) (Fig 1) were supplied by PharmaMar S.A. Mass spectrometry grade acetonitrile, acetic acid, ammonium acetate and formic acid were purchased from ThermoFisher Scientific (Waltham, MA, USA), as was high-performance liquid chromatography grade methyl tert-butyl ether (MTBE). Doxorubicin, doxorubicinol, capecitabine, 5′-deoxyfluorouridine, 5-fluorouracil, α-fluoro-β-alanine, paclitaxel, cisplatin, gemcitabine, 2′,2′-difluoro-2′-deoxyuridine, olaparib, itraconazole, dexamethasone, ondansetron, metoclopramide, bosentan, dimethyl sulfoxide (DMSO) and ammonium hydroxide were purchased from Sigma-Aldrich (St Louis, MO, USA). Acetone was obtained from Merck (Darmstadt, Germany). Ultrapure water was prepared in-house using a Millipore system.

The chemical synthesis of the reference standard for M4 was not feasible. The compound showed only enough stability to be detected and identified by LC-MS/MS but was too unstable to be isolated through conventional methods (e.g., column chromatography on silica gel or reversed-phase chromatography) or characterized by nuclear magnetic resonance spectroscopy. Therefore, M4 and M4-d$_4$ solutions were biosynthetically obtained following the incubation of lurbinectedin and PM040038, respectively, in liver microsomes according to a three-step method described elsewhere [9].

Isolute SLE+ (supported liquid extraction) 200-mg 96-well extraction plates were obtained from Biotage (Uppsala, Sweden), and blank human plasma was obtained from Sera Laboratories International Ltd (West Sussex, United Kingdom) and Scipac Ltd (Sittingbourne, United Kingdom).

RED disposable inserts and base plate were obtained from ThermoFisher Scientific; Sigmacote®, and human recombinant AAG, human recombinant albumin and PBS tablets were obtain from Sigma-Aldrich.

### 2.2 Stock and reference solutions preparation

Stock solutions of lurbinectedin (1 mg/mL) were prepared in DMSO. This was then used to prepare working standard solutions in DMSO:ammonium acetate 10 mM ~pH 3 (1:1, v:v), and the lurbinectedin calibration standard (CS) samples in matrix (0.1–50 ng/mL) were prepared by mixing 10 μL of the corresponding lurbinectedin working standard solution with 490 μL of matrix. Quality control (QC) samples in matrix (50, 40, 3, 0.3 and 0.1 ng/mL) were prepared in a similar way from the QC working solutions. The CS samples in matrix were always freshly prepared. The QC samples were stored at -80˚C ±10˚C and were assayed following one freeze-thaw cycle unless being employed for a specific stability test.

A stock solution of lurbinectedin internal standard (IS) PM040038 (1 mg/mL) (Fig 1) was prepared in DMSO. An intermediate IS solution of 10 μg/mL PM040038 in DMSO:ammonium acetate 10 mM ~pH 3 (1:1, v:v) was used to prepare a 20-ng/mL aqueous IS solution just prior to use.

The M4 reference standard was provided as stock solution, which was then used to prepare intermediate solutions (100 and 200 ng/mL in 0.1% formic acid in acetonitrile:water, 30:70, v:v) for the preparation of CS samples in K$_3$-EDTA plasma (0.5 to 20 ng/mL).

A stock solution of M6 (1 mg/mL) was prepared in DMSO and used to prepare working standard solutions in DMSO, which were then used to prepare CS samples in $K_3$-EDTA plasma (0.5 to 20 ng/mL). The QC samples were prepared in a similar way, but from different stock solutions (0.5, 1.5, 10, 16 and 20 ng/mL).

A stock solution of PM190039 (M6 stable isotope–labeled analogue) was prepared in DMSO (1 mg/mL). A PM190039 intermediate IS solution of 1 μg/mL in DMSO was used to prepare a 5 ng/mL IS solution in acetone. The M4 stable isotope–labeled analogue IS was prepared directly from the stock in acetone (5 ng/mL).

Both the QC and standard used for the *in vitro* PPB experiments were fresh, as they were not stored prior to assay.

## 2.3 Sample preparation procedure

**2.3.1 Lurbinectedin (plasma and plasma:PBS).** The sample extraction was performed by adding 100 μL of the CS, QC or test sample, 50 μL of aqueous IS solution and 50 μL of 1 M ammonium hydroxide to a 300-μL 96-well microtiter plate. This was then mixed and transferred to a 200-mg Isolute SLE+ plate. After 5 minutes of equilibrium, the samples were then eluted from the SLE plate with 1 mL of MTBE into glass inserts and dried under nitrogen. The samples were then reconstituted with 40 μL of acetonitrile:water (1:1, v:v) with 0.1% of formic acid prior to analysis.

**2.3.2 Lurbinectedin metabolites M4 and M6.** The sample extraction was performed by adding 100 μL of the CS, QC or test sample and 100 μL of IS solution to a 2-mL 96-well microtiter plate; 1 mL of MTBE was then added, and the plate mixed for 10 minutes. The plate was then centrifuged at 4°C at 1450 *g* for 10 minutes, after which the aqueous phase was frozen by placing the plate on dry ice. When the aqueous phase was frozen, the MTBE was transferred to a clean 2-mL plate and dried under nitrogen. The samples were then reconstituted with 100 μL of 0.1% formic acid in acetonitrile:water (30:70, v:v) prior to analysis.

## 2.4 Equipment and chromatographic conditions

**2.4.1 Lurbinectedin (plasma and plasma:PBS).** The ultra-performance liquid chromatography (UPLC) system was coupled to a triple quadrupole mass spectrometer. The UPLC system was a Acquity system (Waters, Milford, MA, USA) and composed of a binary pump, column oven and cooled auto sampler. The mass spectrometer was a TQD with an electrospray ionization source from Waters.

The analytical column was a BEH $C_{18}$ (1.7 μm, 50 × 2.1 mm) and was used in conjunction with a pre-column frit (0.2 μm, 2.1 mm) from Waters. The chromatographic separation was performed at 50°C with a flow rate of 0.4 mL/minute utilizing a linear gradient of water (A) and acetonitrile (B), both with 0.1% ammonium hydroxide: 10% to 100% B from 0.00 to 1.00 minute; isocratic at 100% B for 1.50 minute, then returning to initial conditions from 2.50 to 2.51 minutes with a total run time of 3.5 minutes. The samples were kept at 10°C until injection (10 μL).

Data acquisition and processing was performed using Masslynx 4.1 (Waters). The detection was by multiple reaction monitoring (MRM), and the parameters are described in Table 1.

**2.4.2 Lurbinectedin metabolites M4 and M6.** The UPLC system was an Agilent 1290 (Agilent Technologies, Santa Clara, CA, USA) and composed of a binary pump, column oven and cooled auto sampler. The mass spectrometer was an Agilent 6495 with an electrospray ionization source.

The analytical column was a Kinetex® Biphenyl (1.7 μm, 100 × 2.1 mm) from Phenomenex (Torrance, CA, USA). The chromatographic separation was performed at 50°C with a flow rate of 0.4 mL/minute utilizing a linear gradient of water (A) and acetonitrile (B), both with

**Table 1. MS detector parameters.**

| Scan mode | MRM, ESI positive | |
|---|---|---|
| **Lurbinectedin (PM01183)** | MRM transition | 767.7 to 273.0 |
| | Collision energy | 45 eV |
| **Lurbinectedin IS (PM040038)** | MRM parameters | $771.7 \rightarrow 277.1$ |
| | Collision energy | 40 eV |
| **1',3'-dihydroxy-lurbinectedin (M4)** | MRM parameters | $755.3 \rightarrow 481.2$ |
| | Collision energy | 30 eV |
| **1',3'-dihydroxy-lurbinectedin-d$_4$ (M4-d$_4$)** | MRM parameters | $759.3 \rightarrow 481.2$ |
| | Collision energy | 30 eV |
| **N-desmethyl-lurbinectedin (PM030047, M6)** | MRM parameters | $753.3 \rightarrow 479.4$ |
| | Collision energy | 30 eV |
| **N-desmethyl-lurbinectedin-d$_4$ (PM190539)** | MRM parameters | $757.3 \rightarrow 479.2$ |
| | Collision energy | 30 eV |

ESI, positive mode electrospray ionization; IS, internal standard; MRM, multiple reaction monitoring; MS, mass spectrometry.

An incubating microplate shaker from VWR (Radnor, PA, USA) was used during the *in vitro* PPB experiments.

0.1% formic acid: 30 to 100% B from 0.00 to 3.00 minutes; isocratic at 100% B for 1 minute, then returning to initial conditions from 4.00 to 4.1 minutes with a total run time of 5 minutes. The samples were kept at 10°C until injection (10 μL).

Data acquisition and processing was performed using MassHunter software v B.08.01 (Agilent Technologies). The detection was by MRM; the parameters are described in Table 1.

## 2.5 Analytical methodology of PPB

RED was considered the best alternative over other methods (e.g., ultrafiltration) to separate lurbinectedin unbound fraction (*fu*), due to high non-specific binding (NSB) of lurbinectedin leading to adsorption. The RED device consists of a cylinder of dialysis membrane with an 8-kDa molecular weight cut-off held in place by a polytetrafluoroethylene support; the dialysis membrane tube is closed at one end. The RED devices are disposable and used in a reusable polytetrafluoroethylene support plate (holds a maximum of 48 devices). The design of the plate and device allows for PBS to be added in such a way that it can circulate freely around the cylinder of dialysis membrane. In use, spiked plasma (400 μL) is added to the dialysis membrane cylinder part of the RED device, and PBS (600 μL) is added to the other side. The plate, device and volumes used result in a consistent fluid level across both sides of the membrane, preventing mass movement of fluid across the membrane due to hydrostatic pressure.

Each measurement was performed using three RED devices; the devices were placed in the plate and spiked plasma and PBS were added. The plate was then mixed on an incubating microplate shaker at 100 rpm and 37°C for 8 hours.

Both the plasma (diluted 1 in 20 with plasma:PBS 1:1, v:v) and PBS (diluted with plasma 1:1, v:v) were assayed in triplicate using the extraction and UPLC-MS/MS method described above. Transfer of the PBS from the incubation plate into an equivalent volume of plasma prior to assay was performed using pipette tips that had been pre-treated with Sigmacote®.

## 2.6 Method validation

A complete validation of the bioanalytical assays for lurbinectidin and metabolites M4 and M6 in K$_3$-EDTA plasma, and lurbinectedin in K$_3$-EDTA plasma:PBS, was performed according to

the regulatory guidelines [10,11] in terms of selectivity, matrix effect, lower limit of quantitation (LLOQ), linear calibration range, accuracy and precision, whole-blood stability, bench top stability and long-term sample storage.

The calibration curves were fitted by linear regression using a $1/\chi^2$ weighting. The acceptance criterion required that the back-calculated concentrations of the CS be within ±15% of the nominal value (±20% for the LLOQ) for ≥75% of the CS.

The intra-day precision and accuracy of the methods was evaluated on three different analytical days by analyzing six replicates of the five different concentration levels of QC samples. The results of the three analytical batches were also used to evaluate the inter-day precision and accuracy. The inter- and intra-day precision was evaluated by calculating the percent coefficient of variation (%CV) of the back-calculated values at each level. The acceptance criterion was defined as a %CV of ≤15% (≤20% at the LLOQ).

Inter- and intra-day accuracies were calculated as the percentage difference of the measured concentration from the nominal concentration (bias). The acceptance criterion required that the mean back-calculated value of the six replicates have a bias of ±15% from the nominal concentration (±20% for the LLOQ).

The suitability of a 20-fold (for lurbinectedin) and 5-fold (for the metabolites) dilution was assessed by diluting a QC sample with blank matrix.

The selectivity of the lurbinectedin method in plasma was assessed with and without the addition of IS, using blank matrix from six different sources, including hemolyzed, and hyperlipidemic $K_3$-EDTA plasma. The selectivity of the lurbinectedin method in plasma was also assessed using blank matrix spiked with the following co-medications (and selected metabolites of the co-medications): doxorubicin (2 μg/mL), doxorubicinol (0.1 μg/mL), capecitabine (2.5 μg/mL), 5′-deoxyfluorouridine (4 μg/mL), 5-fluorouracil (0.1 μg/mL), α-fluoro-β-alanine (4 μg/mL), paclitaxel (2 μg/mL), cisplatin (15 μg/mL), gemcitabine (15 μg/mL), 2′,2′-difluoro-2′-deoxyuridine (40 μg/mL) and olaparib (5 μg/mL). The selectivity of the lurbinectedin metabolites method was assessed in matrix containing itraconazole (1 μg/mL), dexamethasone (100 ng/mL), ondansetron (300 ng/mL), metoclopramide (25 ng/mL) and bosentan (1 μg/mL).

The criterion for acceptance required that any interfering peak be <20% of the peak area of the analyte at the LLOQ, analyzed within the same run. The response of the IS peak area was considered acceptable when it was <5% of the mean IS peak area at the LLOQ level.

The matrix effect on ionization was assessed by extracting blank $K_3$-EDTA plasma from six different sources including hemolyzed, hyperlipidemic plasma. The plasma extract was then spiked with the analyte and IS at the lower QC (LQC) and higher QC (HQC) levels (assuming 100% recovery); replicates of the spiking solution were also injected. This was also performed using plasma containing the same co-medications and metabolites used in the selectivity experiments.

The matrix factor (MF) for both the analyte and IS was calculated by taking the peak areas found in the presence of matrix and dividing by the mean peak area found in the spiking solution. The IS-normalized MF was calculated by dividing the MF of the analyte by the MF of the IS. The criterion for acceptance was a %CV of the IS-normalized MF of ≤15%.

The carry-over was assessed in each "accuracy and precision" batch by injecting four blank plasma samples after the injection of the highest CS. The area responses in the blank samples were compared to the mean area response of the LLOQ in the same batch. For acceptance, carry-over in the blank sample following the high concentration standard should not be >20% of the LLOQ QC samples in the same batch and 5% for the IS.

The stability of stock and working solutions, as well the stability of IS stock solution, following storage at -20˚C ±5˚C were investigated by concordance verification by comparing the stored solution against a freshly prepared one.

To ensure correct sample collection during the clinical trial procedures, whole-blood (using $K_3$-EDTA as anticoagulant) stability was studied by spiking the blood (lurbinectedin at 20 ng/mL and M4 and M6 at 10 ng/mL). Aliquots of the whole blood were then centrifuged at 4˚C for 0, 30, 60, 90 and 120 minutes post equilibration (15 minutes at 37˚C) to produce plasma, which was then frozen prior to analysis. This was performed at room temperature (and in an ice bath for lurbinectedin). The derived plasma samples were analyzed in triplicate, and the criterion for acceptance was defined as having a response within ±15% of the zero-minute sample.

Likewise, a wide range of long-term stability tests reflecting the sort of real-world storage issues that can occur in hospital-based clinical studies (e.g., short-term storage at -20˚C before transfer to -80˚C, availability of -60˚C but not -80˚C), was evaluated to ensure plasma samples are analyzed before degradation can occur. Therefore, the following plasma stability tests were performed: room temperature, freeze thaw (one cycle consisted of 2 hours at room temperature followed by freezing for at least 12 hours at -80˚C ± 10˚C) and long-term (at -20˚C ± 5˚C, -60˚C ± 10˚C and -80˚C ± 10˚C). For each stability test, two concentrations (LQC and HQC) and six replicates per level were analyzed. For acceptance criteria, the precision (%CV) was ≤15% and the bias when compared with the nominal concentration was ±15%.

The stability in processed samples was assessed by re-injection of the CS and QC samples from an accuracy and precision batch after storage at nominally 10˚C for 3 days. Processed samples were considered suitable for storage and re-analysis if the QC samples passed normal acceptance criteria, and the intra-batch precision was ≤15% (%CV) and the bias was ±15% of the nominal concentration.

As part of the validation of the PPB methodology, the stability of lurbinectedin in $K_3$-EDTA plasma at 37˚C was assessed for 8 hours to confirm the PPB method was not impacted by any instability of the analyte. This analysis was performed at the LQC and HQC levels and was assayed using the normal plasma assay.

The recovery was evaluated by processing six replicates of the LQC, middle QC (MQC) and HQC samples and blank matrix samples. The blank matrix samples were then spiked with the analyte and IS assuming 100% recovery (six at each level). The response of the analyte and IS was then compared to the mean response found in the spiked blank samples (at each level).

## 2.7 *In vitro* PPB experiments

*In vitro* PPB experiments to determine lurbinectedin *fu* were conducted with RED in $K_3$-EDTA plasma from donors. Fresh plasma (not previously frozen), previously frozen plasma (fresh plasma subjected to four cycles of freeze [at -80˚C ±10˚C for ≥12 hours] then thawed [at room temperature for 2 hours]) and diluted fresh plasma (not previously frozen, diluted 1:1 with PBS prior to incubation) were spiked to 100 ng/mL with lurbinectedin and then equilibrated with PBS in RED devices for 8 hours at 37˚C and assayed.

PPB of lurbinectedin was also studied in artificial plasma (PBS with recombinant human AAG and albumin) at different levels of AAG, albumin and lurbinectedin to estimate the dissociation rate constants for each of these plasma proteins. Several experimental conditions, performed in triplicate, with triplicate analyses, were explored:

- Condition 1 with lurbinectedin at 500 ng/mL, AAG at 0, 0.25, 0.75, 1, 2 and 4 mg/mL and albumin at 40 mg/mL.

- Condition 2 with lurbinectedin at 500 ng/mL, AAG at 1 mg/mL and albumin at 25, 35, 40, 45 and 50 mg/mL.

- Condition 3 with lurbinectedin at 50, 75, 100, 250 and 500 ng/mL, AAG at 1 mg/mL and albumin at 40 mg/mL.

In addition, the stability of lurbinectedin PPB in fresh plasma following storage at -80˚C ±10˚C was evaluated; aliquots of fresh plasma samples (not previously frozen) from three donors were spiked with 500 ng/mL of lurbinectedin. The samples were stored for at least 21, 30, 62 and 91 days prior to analysis; the day 0 aliquots were not frozen before analysis. The spiked aliquots from the three donors were equilibrated with PBS in RED devices in triplicate. The samples generated in the RED devices were analyzed in triplicate.

## 2.8 Method application

The method for lurbinectedin in plasma was developed to be used in clinical studies with lurbinectedin.

The method for lurbinectedin in plasma:PBS and the dissociation rate constants calculated with the *in vitro* PBS experiments with artificial plasma were used to derive the unbound concentration of lurbinectedin.

The metabolites methods were used to further determine their proportion of main lurbinectedin metabolites with respect to the parent compound and how this proportion can be affected by different conditions, such as the concomitant administration of CYP3A4 inhibitors and inducers, and different degrees of hepatic impairment.

## 3 Results and discussion

### 3.1 Method development

The method for the quantification of lurbinectedin concentration in plasma was based on that previously reported for pre-clinical samples [5], but several improvements were introduced to reduce the run time analysis to 3.5 minutes, which is critical for its use in large clinical trials. One of the main changes was the use of 0.1% ammonium hydroxide as the mobile phase modifier. This change from 0.1% formic acid decreased the retention of lurbinectedin and the peak width, making the peak height greater and thereby increasing the sensitivity. Additionally, the method was selective for lurbinectedin, even in the presence of co-medications. No matrix effect was detected. However, some carry-over was observed.

During the method development for the determination of lurbinectedin PPB in $K_3$-EDTA plasma:PBS (1:1, v:v) using RED devices, it became apparent that in the absence of plasma proteins, lurbinectedin showed very high levels of NSB. To counteract the NSB, Sigmacote®-treated pipette tips were used.

For the quantitation of the two main lurbinectedin metabolites in human plasma in dedicated clinical pharmacology studies, new methods were developed as the gradient phase chromatography of the lurbinectedin method did not allow the inclusion of these metabolites. Moreover, a single method for both metabolites was the preferred option. Hence, the chromatography method was developed so that baseline separation was achieved for both metabolites and parent compound. This was achieved by changing the modifier to 0.1% formic acid, which increased the retention of the compounds by the column, combined with an optimized elution gradient.

The M4 and M4-$d_4$ solutions were biosynthetically obtained. Since chemically synthesized M6 and M6-$d_4$ were available, the decision to perform two identical assays, although with different CS and QC samples for each metabolite, was taken in this case. Nevertheless, the approach of using biologically generated and adequately quantified metabolites (and corresponding IS) is certainly an adequate option in the study of metabolite PK profiles.

## 3.2 Method validation

### 3.2.1 Lurbinectedin in plasma

The calibration curves for lurbinectedin displayed good linearity over the concentration range of 0.1 to 50 ng/mL. A linear regression equation with a weighting factor of $1/\chi^2$ for the analyte IS ratio was found to produce the best fit for the concentration-response relationship.

The intra- and inter-day precisions for the quantification of lurbinectedin ranged from 2.7% to 12.9% and from 5.1% to 10.7%, respectively. Similarly, the within- and between-day accuracy (bias) ranged from -10% to 12% and -5% to 6%, respectively. For the LLOQ, the intra-day precisions ranged from 11.9% to 15.4% and the inter-day precision was 15.8%; the intra-day accuracy (bias) ranged from -1% to 17% and the inter-day accuracy (bias) was 5%. The results are summarized in Table 2.

The ability to perform adequate determinations after a 20-fold dilution was also demonstrated, the accuracy (bias) and precision were -5.0% and 3.6%, respectively.

Selectivity for lurbinectedin in different source matrices, including those containing co-medications, was demonstrated; an example of a blank sample with IS from the selectivity assay is shown in Fig 2. Selectivity for the IS was also demonstrated; an example of an LLOQ sample with IS is shown in Fig 2.

The matrix effect on ionization for lurbinectedin was evaluated using the precision (%CV) of the IS-normalized MF, this was found to be 4.5% and 3.9% for the LQC and HQC levels, respectively. The precision (%CV) of the IS-normalized MF in the presence of co-medication and metabolites was 5.6% and 2.9% for at the LQC and HQC levels, respectively.

In the carry-over evaluation, an analyte peak >20% of the response of the LLOQ level was found in the first of the four blank samples following the upper limit of quantitation (ULOQ) sample; the following samples were clean. In view of this, the injection of one "zero" sample (processed blank plasma without analyte but with IS) before samples with expected low values that follow high-value samples is required during the quantification of clinical samples.

The stability of lurbinectedin in DMSO for ≥32 days at -20˚C ±5˚C was demonstrated. Similarly, the stability of the IS (PM040038) stock solution in DMSO was demonstrated for ≥36 days at -20˚C ±5˚C. However, the stability of the lurbinectedin working solutions (5 and 2500 ng/mL) in DMSO:ammonium acetate 10 mM ~pH 3 (1:1, v:v) following storage for 7 days at -20˚C ±5˚C was not demonstrated for the 5 ng/mL solution; therefore, the spiking solutions need to be prepared on the day of use.

The results of the whole-blood stability study at room temperature showed a difference of -5.1% in the lurbinectedin concentration from time zero to 120 minutes, while at 4˚C ±2˚C, the observed difference from time zero to 120 minute was -8.5%. Therefore, the stability of lurbinectedin in human whole $K_3$-EDTA blood for ≥120 minutes was demonstrated.

The stability of lurbinectedin in $K_3$-EDTA plasma was assessed using LQC and HQC samples. The results are summarized in Table 3. In all cases, the mean concentration at each level was within ±15% of the nominal concentration and the %CV was ≤15%. Of note, stability of 853 days at -80˚C was demonstrated.

Processed samples were found to be stable when reinjected following storage for 3 days at nominally 10˚C. The mean bias was -1%, 3% and 2% for the LQC, MQC and HQC samples, respectively.

### 3.2.2 Lurbinectedin in plasma:PBS.
The method for the determination of lurbinectedin in $K_3$-EDTA plasma:PBS (1:1, v:v) for the determination of its PPB was performed in a very similar way to the plasma-only assay. The calibration curves displayed good linearity with a weighting factor of $1/\chi^2$.

**Table 2. Assay performance for the analysis of lurbinectedin in plasma, in plasma:PBS and its metabolites.**

| Performance | Nom. conc. (ng/mL) | Measured conc. (ng/mL) | Bias (%) | CV (%) |
|---|---|---|---|---|
| **Lurbinectedin plasma** | | | | |
| Intra-day | 0.1 | 0.117 | 17 | 15.4 |
| | 0.3 | 0.336 | 12 | 12.9 |
| | 3 | 2.88 | -10 | 7.4 |
| | 40 | 40.4 | 9 | 6.3 |
| | 60 | 50.5 | 7.5 | 7.5 |
| Inter-day | 0.1 | 0.105 | 5 | 15.8 |
| | 0.3 | 0.311 | 4 | 10.7 |
| | 3 | 2.84 | -5 | 6.2 |
| | 40 | 42.3 | 6 | 6.3 |
| | 60 | 49.3 | -1 | 5.1 |
| **Lurbinectedin plasma:PBS** | | | | |
| Intra-day | 0.1 | 0.111 | 20 | 10.6 |
| | 0.3 | 0.315 | 5 | 9.2 |
| | 3 | 2.96 | 7 | 4.9 |
| | 40 | 41.1 | 7 | 3.9 |
| | 50 | 51.2 | 8 | 4.2 |
| Inter-day | 0.1 | 0.112 | 12 | 9.0 |
| | 0.3 | 0.312 | 4 | 6.6 |
| | 3 | 3.06 | 2 | 5.2 |
| | 40 | 42.0 | 5 | 3.1 |
| | 50 | 52.9 | 6 | 3.9 |
| **Metabolite 4 plasma** | | | | |
| Intra-day | 0.5 | 0.555 | 15 | 10.2 |
| | 1.5 | 1.50 | 15 | 8.6 |
| | 10 | 10.7 | 7 | 7.2 |
| | 16 | 16.8 | 6 | 4.4 |
| | 20 | 22.2 | 11 | 5.7 |
| Inter-day | 0.5 | 0.529 | 6 | 12.9 |
| | 1.5 | 1.56 | 4 | 9.8 |
| | 10 | 10.6 | 6 | 5.4 |
| | 16 | 16.6 | 4 | 4.5 |
| | 20 | 21.7 | 9 | 5.7 |
| **Metabolite 6 plasma** | | | | |
| Intra-day | 0.5 | 0.534 | 19 | 14.9 |
| | 1.5 | 1.58 | 14 | 10.2 |
| | 10 | 10.6 | 9 | 13.3 |
| | 16 | 17.0 | 11 | 9.8 |
| | 20 | 22.1 | 11 | 11.1 |
| Inter-day | 0.5 | 0.562 | 12 | 10.5 |
| | 1.5 | 1.65 | 10 | 8.2 |
| | 10 | 10.6 | 6 | 9.0 |
| | 16 | 17.1 | 7 | 7.6 |
| | 20 | 21.3 | 7 | 7.5 |

conc., concentration; CV, coefficient of variation; nom., nominal; PBS, phosphate-buffered saline.

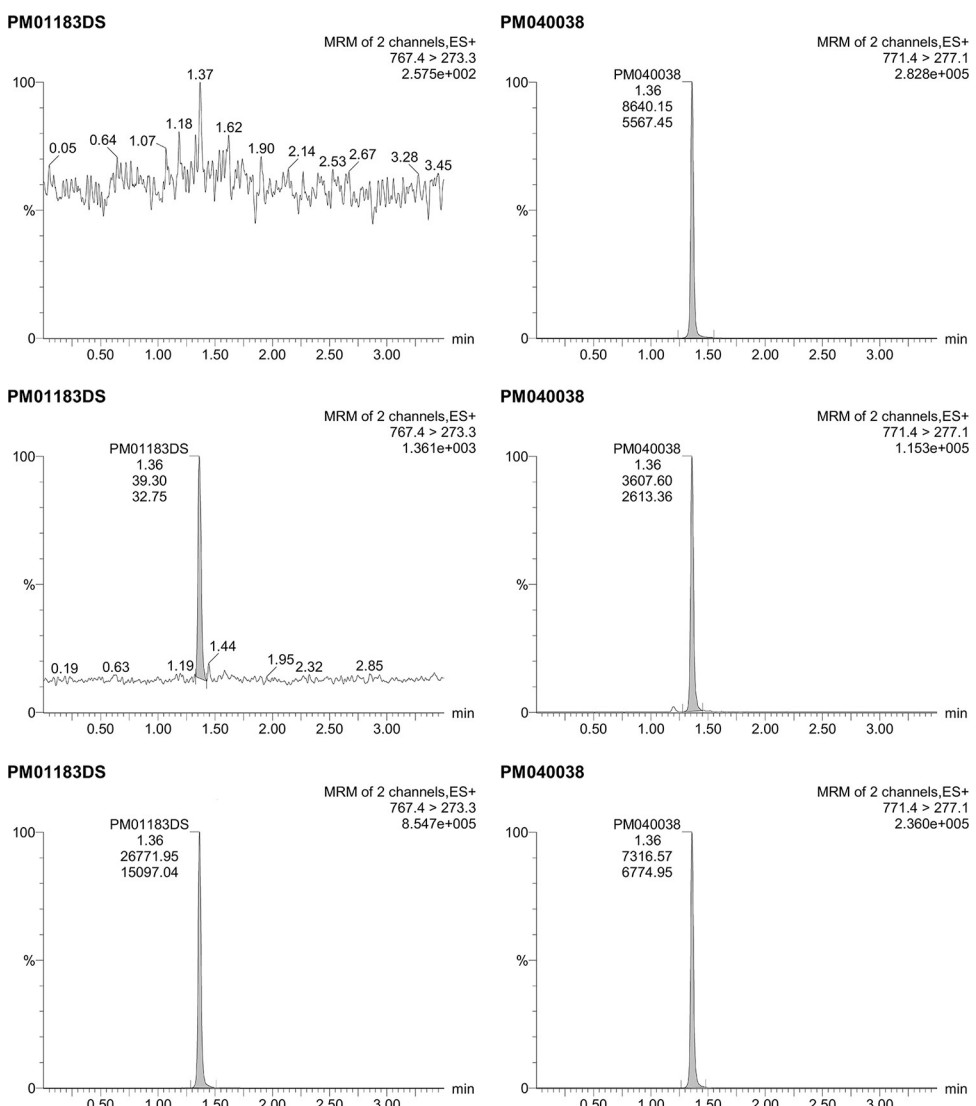

**Fig 2. Chromatogram of Lurbinectedin Blank (upper), LLOQ (middle), and a Treated Patient (lower) Plasma Samples (with a lurbinectedin concentration of 177 ng/mL), With Internal Standard (right).** ES+, positive mode electrospray ionization; LLOQ, lower limit of quantitation; MRM, multiple reaction monitoring.

The intra- and inter-day precisions for the quantification of lurbinectedin ranged from 1.4% to 9.2% and from 3.1% to 6.6%, respectively. Similarly, the within- and between-day accuracy (bias) ranged from -1% to 8% and 4% to 6%, respectively. For the LLOQ, the intra-day precisions ranged from 4.2% to 10.6% and the inter-day precision was 9.0%; the intra-day accuracy (bias) ranged from 6% to 20% and the inter-day accuracy (bias) was 12%. The results are summarized in Table 2.

The ability to perform adequate determinations after a 20-fold dilution was demonstrated by the accuracy (bias) and precision, which were 7.2% and 7.3%, respectively.

In the carry-over evaluation, an analyte peak with >20% of the response of the LLOQ level was found in only one of the first of the four blank samples following the ULOQ sample; all other samples were clean (no peaks were found for the IS; Fig 3). In view of this, the injection of one "zero" sample (processed blank plasma without analyte but with IS) before samples

**Table 3. Stability of lurbinectedin and its metabolites M4 and M6 in human K$_3$-EDTA plasma under different test conditions.**

| Stability test | Nom. conc. (ng/mL) | Measured conc. (ng/mL) | Bias (%) | CV (%) |
|---|---|---|---|---|
| **Lurbinectedin plasma** | | | | |
| Ambient, 16.5 hours | 0.3 | 0.276 | -8 | 6.2 |
| | 40 | 41.0 | 3 | 7.8 |
| 4 freeze-thaw cycles | 0.3 | 0.289 | -4 | 4.9 |
| | 40 | 38.0 | -5 | 4.4 |
| -20˚C ±5˚C, 23 days | 0.3 | 0.271 | -10 | 8.0 |
| | 40 | 36.2 | -9 | 4.1 |
| -60˚C ±10˚C, 245 days | 0.3 | 0.285 | -5 | 10.0 |
| | 40 | 36.1 | -10 | 2.5 |
| -80˚C ±10˚C, 853 days | 0.3 | 0.269 | -10 | 11.9 |
| | 40 | 36.3 | -9 | 2.6 |
| 37˚C, 8 hours | 0.3 | 0.293 | -2 | 3.5 |
| | 40 | 40.2 | 1 | 6.0 |
| **Metabolite 4 plasma** | | | | |
| Ambient, 6 hours | 1.5 | 1.40 | -7 | 7.7 |
| | 16 | 13.7 | -14 | 6.9 |
| 5 freeze-thaw cycles | 1.5 | 1.58 | 5 | 6.0 |
| | 16 | 15.6 | -3 | 5.0 |
| -80˚C ±10˚C, 197 days | 1.5 | 1.54 | 3 | 6.9 |
| | 16 | 15.8 | -1 | 5.1 |
| **Metabolite 6 plasma** | | | | |
| Ambient, 21.8 hours | 1.5 | 1.58 | 5 | 7.7 |
| | 16 | 16.8 | 5 | 12.7 |
| 5 freeze-thaw cycles | 1.5 | 1.69 | 13 | 9.2 |
| | 16 | 18.2 | 14 | 2.9 |
| -80˚C ±10˚C, 237 days | 1.5 | 1.70 | 13 | 6.8 |
| | 16 | 18.4 | 15 | 5.4 |

conc., concentration; CV, coefficient of variation; nom., nominal; PBS, phosphate-buffered saline.

with expected low values that follow high value samples is required during the quantification of samples.

The recovery of lurbinectedin in K$_3$-EDTA plasma:PBS (1:1, v:v) was evaluated and found to be in the range of 66.4% to 86.6% (analyte and IS). When the recovery was normalized using the IS, the normalized recovery had a %CV of 6.2% across all three levels.

The PPB of lurbinectedin remained very high (>99%) following storage at -80˚C ±10˚C for ≥91 days. There was no difference in the PPB between day 0 (fresh plasma) and day 91 (after storage at -80˚C ±10˚C).

**3.2.3 Lurbinectedin metabolites.** The calibration curves for lurbinectedin metabolites displayed good linearity over the concentration range of 0.5 to 20 ng/mL. A regression equation with a weighting factor of $1/\chi^2$ for the analyte IS ratio was found to produce the best fit for the concentration response relationship.

The intra- and inter-day precisions for the quantification of M4 ranged from 2.3% to 10.2% and from 4.5% to 12.9%, respectively. Similarly, the within- and between-day accuracy (bias) ranged from -9% to 15% and 4% to 9%, respectively. For M6, the intra- and inter-day precisions ranged from 3.0% to 14.9% and from 7.5% to 10.5%, respectively, with the within-

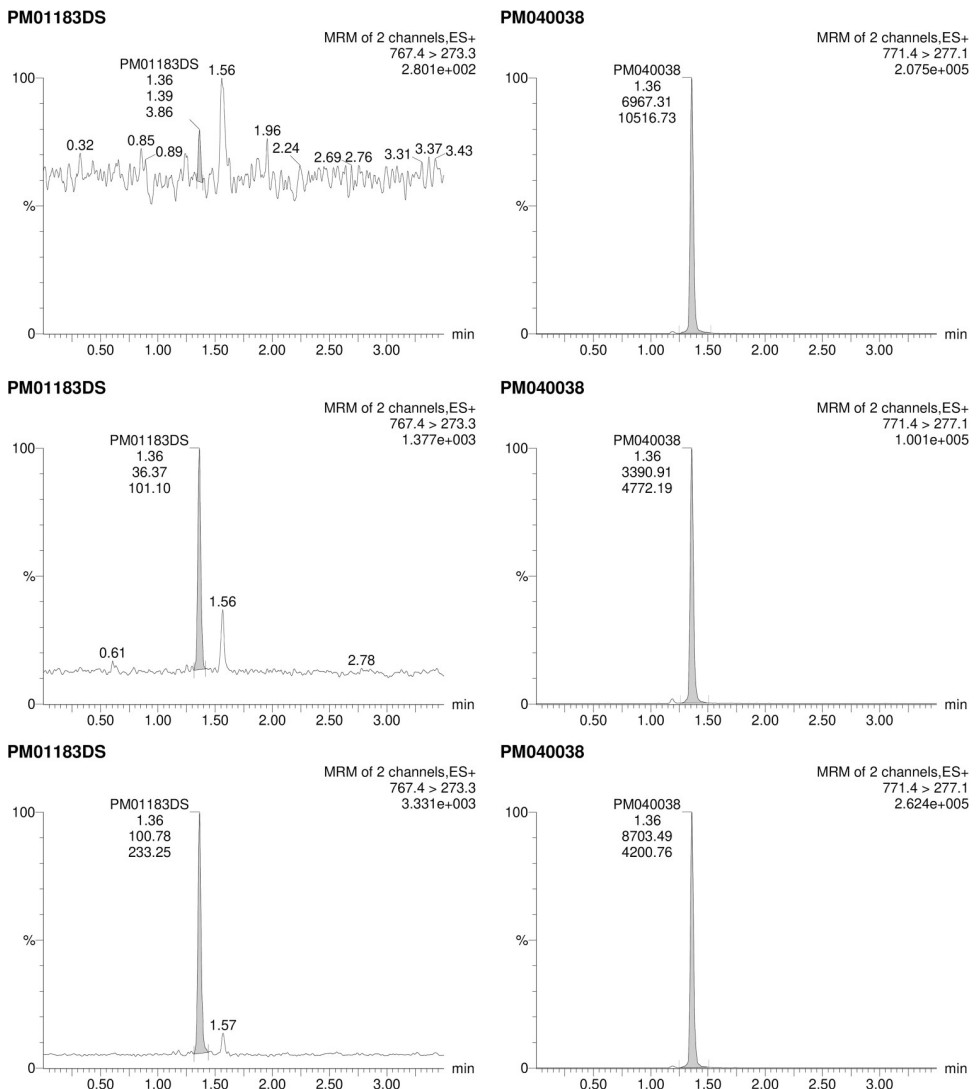

**Fig 3. Chromatogram of Lurbinectedin Blank (upper), LLOQ (middle), and a Treated Patient Plasma:PBS Samples, With Internal Standard (right).** LLOQ, lower limit of quantitation; MRM, multiple reaction monitoring; PBS, phosphate-buffered saline.

and between-day accuracy (bias) ranging from 1% to 14% (19% at the LLOQ) and 6% to 12%, respectively. The results are summarized in Table 2.

The ability to perform adequate determinations after a 5-fold dilution was also demonstrated, the accuracy (bias) and precision for M4 were -2% and 9.8%, respectively, and for M6 were 11.9% and 7.6%.

Selectivity with and without IS in different source matrices including those containing co-medications was demonstrated for both M4 and M6; examples of blank samples with IS from the selectivity assay are shown in Figs 4 and 5, respectively. Selectivity for the M4 and M6 IS was also demonstrated; examples of LLOQ samples with IS and treated patient samples are also shown in Figs 4 and 5.

The matrix effect on ionization for M4 was evaluated using the precision (%CV) of the IS-normalized MF; this was found to be 6.5% and 4.3% for the LQC and HQC samples,

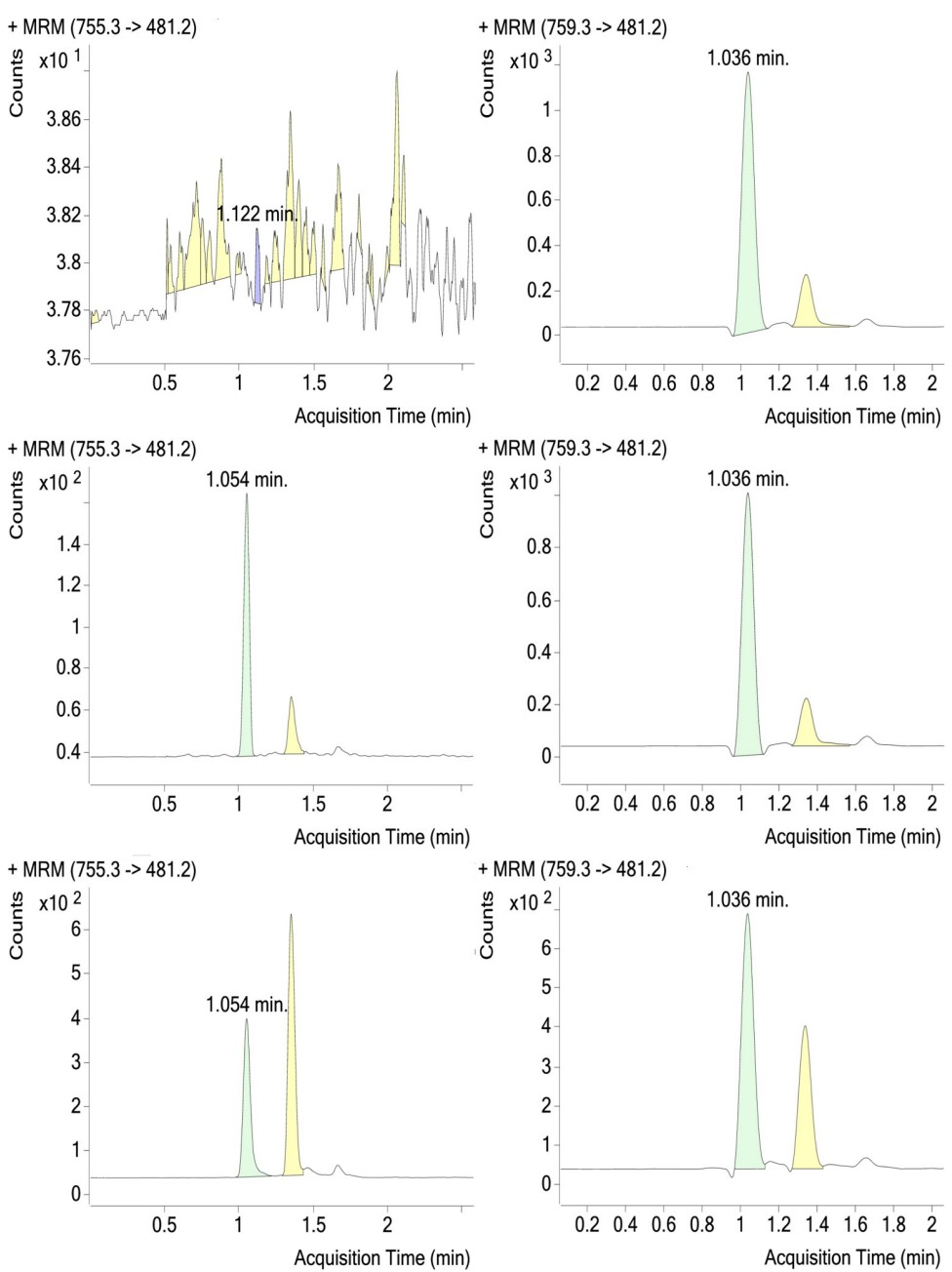

**Fig 4. Chromatogram of M4 Blank (upper), LLOQ (middle), and a Treated Patient (lower) Plasma Samples, With Internal Standard (right).** Retention time of M4 is ~1.05 minutes. LLOQ, lower limit of quantitation; MRM, multiple reaction monitoring.

respectively. Likewise, the precision (%CV) of the IS-normalized MF for M6 was found to be 5.2% and 4.1% for the LQC and HQC samples.

In the carry-over evaluation, no peaks >20% of the response of the LLOQ level were found in any of the blank samples following the ULOQ samples for both metabolites.

The stability of the M6 and PM190539 (IS) stock solutions in DMSO (1 mg/mL) for ≥28 days at -20˚C ±5˚C was demonstrated.

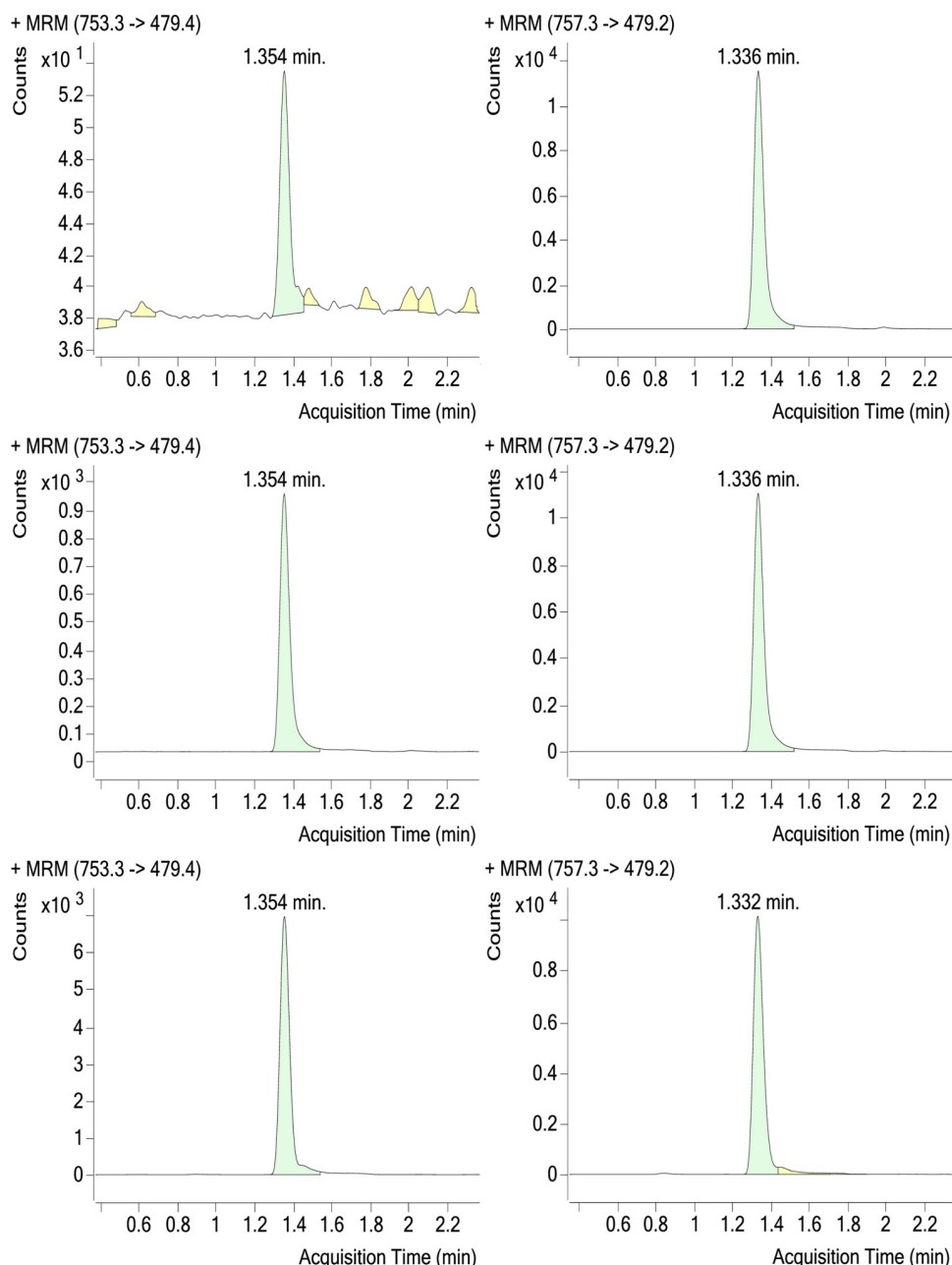

**Fig 5. Chromatogram of M6 Blank (upper), LLOQ (middle), and a Treated Patient (lower) Plasma Samples, With Internal Standard (right).** LLOQ, lower limit of quantitation; MRM, multiple reaction monitoring.

The results of the whole-blood stability study at room temperature showed a difference of -3% in the M4 concentration from time zero to 120 minutes, while the difference for M6 was -8%. Therefore, the stability of metabolites in human whole $K_3$-EDTA blood for $\geq$120 minutes was demonstrated.

The stability of lurbinectedin in $K_3$-EDTA plasma was assessed using LQC and HQC samples. The results are summarized in Table 3. In all cases, the mean concentration at each level was within ±15% of the nominal concentration and the %CV was $\leq$15%.

Processed samples were found to be stable when reinjected following storage for 91 hours at nominally 10˚C.

### 3.3 *In vitro* PPB experiments

The mean percent unbound lurbinectedin in fresh plasma (0.400%; 99.6% bound) and previously frozen plasma (0.443%; 99.6% bound) from donors were found to be very similar, whereas in diluted fresh plasma (1:1 with PBS) it was found to be reduced (0.668%, 99.3% bound), as anticipated. In fact, this dilution effect has been used in the evaluation of PPB for highly bound drugs [12].

In artificial plasma, lurbinectedin *fu* did not change in the experimental condition 2 (varying albumin concentration) and experimental condition 3 (varying lurbinectedin concentration), with values of 0.404% (99.6% bound) and 0.393% (99.6% bound), respectively.

On the other hand, the mean percent unbound lurbinectedin in the experimental condition 3 (varying AAG concentration) changed from 8.95% (91.1% bound) without AAG to 1.68% (98.3% bound), 0.604% (99.4% bound), 0.473% (99.5% bound) 0.259% (99.7% bound) and 0.101% (99.9% bound) with AAG levels of 0.25, 0.75, 1, 2 and 4 mg/mL, respectively (Fig 6).

The stability of lurbinectedin PPB in fresh plasma from three donors following storage at -80˚C ±10˚C was successfully demonstrated for ≥91 days, as no differences in mean percent unbound lurbinectedin were observed between the fresh plasma (0.151% to 0.335%; 99.8% to 99.7% bound) and day 21 (0.206% to 0.319%; 99.8% to 99.7% bound), day 30 (0.168% to 0.266%; 99.8% to 99.7% bound), day 62 (0.196% to 0.286%; 99.8% to 99.7% bound) and day 91 (0.159% to 0.213%; 99.8% to 99.8% bound) after storage at -80˚C ±10˚C.

The selectivity and matrix effect assays were not performed during the validation in $K_3$-EDTA plasma:PBS (1:1, v:v) because the change in matrix would have only reduced any effect present, as the new matrix is a diluted form of the original matrix.

### 3.4 Method application

The method for lurbinectedin in plasma has been successfully applied in the majority of early and late phase clinical studies, where lurbinectedin was used as single agent or in combination with other chemotherapeutic agents [8].

The method for lurbinectedin in plasma:PBS and the *in vitro* PBS experiments with artificial plasma served to demonstrate the extent to which AAG levels affect lurbinectedin *fu*. At

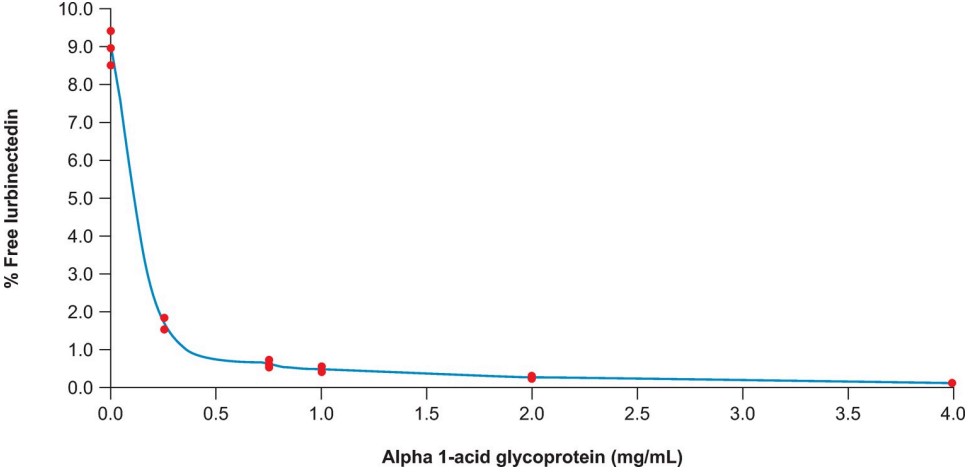

**Fig 6. Lurbinectedin plasma protein binding in artificial plasma with varying alpha-1-acid glycoprotein levels.**

physiological concentrations used (~1 mg/mL AAG, ~50 mg/mL albumin [13]), *fu* was equivalent to that found in fresh plasma. However, when the AAG levels decreased the amount of unbound drug dramatically increased, and when the AAG levels increased the unbound drug mildly decreased. This is relevant, since large changes in AAG have been observed in cancer patients with lung tumors (1.65 ±0.16 mg/mL) and liver cirrhosis (0.67 ±0.05 mg/mL) [13]. Moreover, equilibrium dissociation constants between lurbinectedin and AAG and albumin could be calculated to allow estimating lurbinectedin unbound plasma concentration from individual total plasma concentration, AAG and albumin levels, to be used in exposure-response (E-R) models of safety and efficacy published elsewhere [9]. Interestingly, in E-R analyses with safety outcomes (i.e. incidence of severe neutropenia and thrombocytopenia) involving a large number of patients (n = 644), statistically significant relationships could be established either with total and with unbound lurbinectedin exposure, while in E-R analyses of efficacy (i.e. overall survival and objective response rate), a relationship could be established with unbound lurbinectedin exposure only.

Despite the importance of AAG levels on lurbinectedin unbound plasma concentration, the clinical relevance of potential drug-drug interactions due to protein binding displacement, is deemed not relevant based on the low plasma concentration of lurbinectedin relative to that of AAG.

All of the aforementioned methods were used in clinical pharmacology studies of lurbinectedin, such as PM1183-A-019-21, a two-way crossover, phase 1 drug-drug interaction study that explored the effect of concomitant treatment with the moderate CYP3A4 inducer bosentan on the PK of lurbinectedin in patients with cancer. In all cycles, patients received lurbinectedin at full dose (3.2 mg/m$^2$) through a 1-hour intravenous infusion. Blood samples were taken up to 168 hours post-infusion, collected in vacutainer tubes with $K_3$-EDTA anticoagulant and centrifuged at 4°C for 2000 × g for 10 minutes to separate and store the plasma polypropylene tubes at -70°C until analysis. As an example, Fig 7 shows total lurbinectedin,

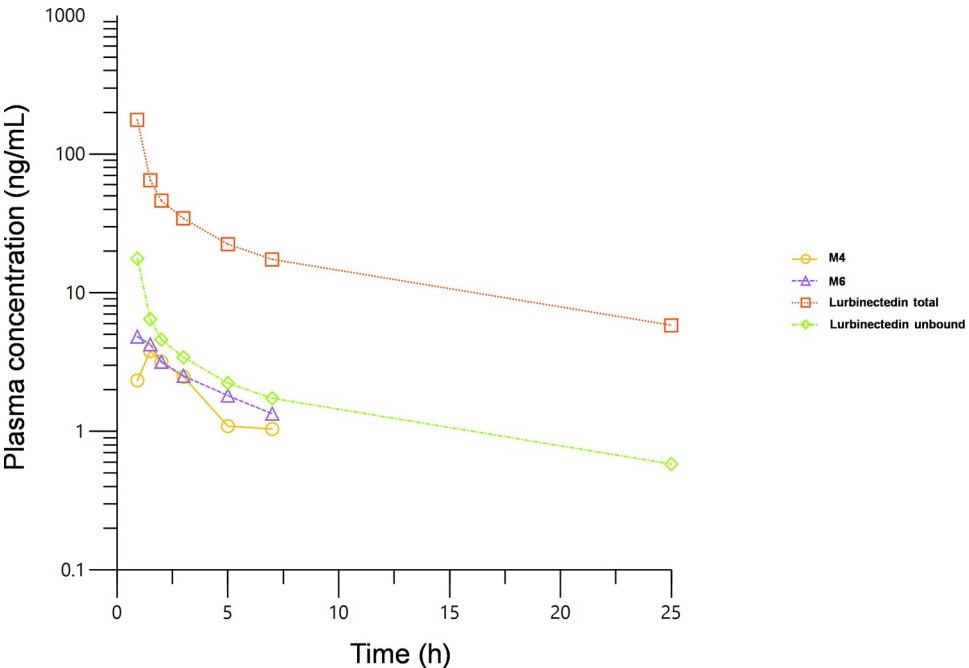

**Fig 7. Total and unbound lurbinectedin and M4 and M6 plasma concentration-time profiles of an individual patient treated in study A-019.**

unbound lurbinectedin (as result of multiplying total lurbinectedin concentration by measured *fu*), and the M4 and M6 plasma concentration-time profiles of an individual patient participating in study PM1183-A-019-21.

## 4 Conclusions

A UPLC–MS/MS method that was rapid and sensitive was developed and validated for the quantitative determination of lurbinectedin in human $K_3$-EDTA plasma. The method displayed good linearity and acceptable within- and between-day precision and accuracy. The long-term stability studies conducted ensure that samples from clinical trials are viable for $\geq$853 days when stored at -80˚C ±10˚C. The UPLC–MS/MS method was then validated in plasma:PBS and applied to donors and artificial plasma samples from PPB RED investigations; lurbinectedin *fu* was found to be affected by neither lurbinectedin concentration, albumin concentration nor storage but was affected by the AAG concentration present in samples.

A method was developed and validated for the quantitative determination of 1',3'-dihydroxy-lurbinectedin and *N*-desmethyl-lurbinectedin in human $K_3$-EDTA plasma. The method displayed good linearity and acceptable within- and between-day precision and accuracy. In the application to the analysis of 1',3'-dihydroxy-lurbinectedin, a biologically derived reference standard was used. The use of such standards could simplify metabolite PK analyses.

The performance data of these methods were demonstrated in clinical samples and meet current guidelines on bioanalytical method validation. The methods proved to be highly selective, specific and sufficiently sensitive for its application in the analysis of clinical samples.

## Acknowledgments

The authors gratefully acknowledge the members of PharmaMar Organic Chemistry and Bioanalysis Laboratories for providing lurbinectedin, 1',3'-Dihydroxy-lurbinectedin, *N*-Desmethyl-lurbinectedin and their deuterated analogues.

## Author Contributions

**Conceptualization:** Nicholas King, Rubin Lubomirov, Raquel Altares, Carlos Fernandez-Teruel, Salvador Fudio.

**Formal analysis:** Nicholas King, Soledad Garcia-Martinez, Eider Alcaraz, Alba Grisalena.

**Methodology:** Nicholas King, Eider Alcaraz, Rubin Lubomirov, Raquel Altares, Carlos Fernandez-Teruel, Salvador Fudio.

**Supervision:** Rubin Lubomirov, Raquel Altares, Carlos Fernandez-Teruel, Andrés M. Francesch, Pablo M. Avilés, Salvador Fudio.

**Validation:** Nicholas King, Soledad Garcia-Martinez, Eider Alcaraz, Alba Grisalena.

**Writing – original draft:** Nicholas King, Salvador Fudio.

**Writing – review & editing:** Rubin Lubomirov, Raquel Altares, Carlos Fernandez-Teruel, Andrés M. Francesch, Pablo M. Avilés, Salvador Fudio.

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
