## [Decision Letter · Decision Letter 0]

15 Oct 2022

PONE-D-22-19338Quantitative determination of lurbinectedin, its unbound fraction and its metabolites in human plasma utilizing ultra-performance LC–MS/MSPLOS ONE

Dear Dr. Nicholas King,

Thank you for submitting your manuscript to PLOS ONE. After careful consideration, we feel that it has merit but does not fully meet PLOS ONE’s publication criteria as it currently stands. Therefore, we invite you to submit a revised version of the manuscript that addresses the points raised during the review process.

We look forward to receiving your revised manuscript.

Kind regards,

Tommaso Lomonaco, Ph.D

Academic Editor

PLOS ONE

Journal Requirements:

Reviewers' comments:

Reviewer's Responses to Questions

**Comments to the Author**

1. Is the manuscript technically sound, and do the data support the conclusions?

Reviewer #1: Yes

Reviewer #2: Yes

2. Has the statistical analysis been performed appropriately and rigorously? 

Reviewer #1: N/A

Reviewer #2: Yes

3. Have the authors made all data underlying the findings in their manuscript fully available?

Reviewer #1: Yes

Reviewer #2: Yes

4. Is the manuscript presented in an intelligible fashion and written in standard English?

Reviewer #1: Yes

Reviewer #2: Yes

5. Review Comments to the Author

Reviewer #1: In general:

The manuscript describes the development and validation of assays for the quantification of lurbinectedin (selective oncogenic transcriptor inhibitor) and its two metabolites M4 and M6 in human plasma (PK) and plasma dialysate (plasma protein binding). For this purpose, two analytical assays based on UPLC coupled to tandem mass spectrometry were developed and validated according to the common FDA and EMA procedures. One assay for the parent drug and an additional method for the two metabolites. In general, appropriate sample preparation and chromatographic/mass spectrometric conditions were applied. This protocol was applied to quantify lurbinectedin in plasma and dialysate samples of clinical study participants (one PK presented).

For the quantification of lurbinectedin in bioanalytical samples only two analytical methods can be found by Pubmed. Both are published by the same group and are mentioned and compared in the introduction. In principle, the presented assays seem good developed, validated, and applied to a clinical trial.

From my point of view and after moderate revision regarding some points described in the details, the manuscript could become acceptable.

In detail:

Abstract

1.) Quantitative results should be mentioned in the abstract. I suggest at least to mention accuracy and precision, linearity, and recovery.

2.) In the abstract, I miss data of an application to real human samples. This should be added.

Introduction

3.) Lines 37-40. Is it really necessary to mention the awkward IUPAC name of the compound? The structure in figure 1 should be sufficient in my opinion.

Materials and methods

4.) Line 113 and chapter 2.6. For what kind of human plasma has the method been developed and validated (K3-EDTA, or else)?

5.) Lines 89 and 231. The references [10,11] are not correct. The guidelines of EMA and FDA are missing in the references list. The others seem to be ok.

6.) Line 325. Again, what kind of plasma. Did you test different anticoagulations?

Results and Discussion

7.) Lines 363-373. Please describe more exactly why the metabolites could not be included in the method for the parent compound.

8.) Figure 2. Please mention the concentration of the patient plasma sample. When comparing the peak areas, it seems as if the patient sample is >50 ng/mL?

9.) Lines 572-581. With regard to the application of the methods to patients, please add a respective chapter in the Materials and Methods part and present at this position (chapter 3.6) only the results of the patient application.

Reviewer #2: This manuscript describes a UPLC-MS/MS assay to determine plasma concentrations of an anticancer drug, lurbinectedin and its two main metabolites. The assay has been adapted from a previously published method in order to shorten the run-time of each sample, allowing faster analysis. This represents a valuable, but not major, improvement. More interestingly, however, this new assay has been applied to a plasma protein binding study. Lurbinectin is highly bound in plasma (unbound fraction [fu] of 0.4%). The unbound fraction is dependent on alpha1—glycoprotein acid plasma levels. The work is original and may be of some benefit for further pharmacokinetic study of this compound.

Specific points:

1. Certain key information is missing on lurbinectidin plasma protein binding: is the unbound fraction independent of plasma concentrations? What is the impact of hypoalbuminemia on fu? Indeed, the authors have evaluated low and high alpha1—glycoprotein acid plasma levels on fu, but abnormal albuminemias occur at least as often.

2. What about the plasma protein binding of the two main metabolites (M4 and M6)?

3. Conclusions. The clinical relevance of DDI corresponding to the plasma protein binding displacement of lurbinectidin —a drug with a rather low hepatic extraction ratio— should be discussed or at least mentioned. Lastly, reflecting the poor development of the Discussion, the list of references is too limited.

4. Introduction (page 4, lines 66-68): It seems that the PK/PD analysis was performed using fu estimated from plasma AAG and albumin levels to correct total lurbinectedin plasma concentrations. The main results should be given somewhere, compared with non-corrected results.

5. The choice of some of the drugs used to evaluate the selectivity of this assay should be justified. If doxorubicin, and its metabolite, is logical considering the combination of lurbinectedin plus doxorubicin —as is the case for antiemetic drugs, e.g.—, that of platinum compounds seems unlikely. Why is itraconazole considered, but not rifampicin?

6. Calibration curves (Page 16): What about the intercept concerning the carry-over phenomenon?

6. PLOS authors have the option to publish the peer review history of their article (what does this mean?). If published, this will include your full peer review and any attached files.

Reviewer #1: **Yes: **Jürgen Burhenne

Reviewer #2: **Yes: **étienne Chatelut

---

## [Author Response · Author response to Decision Letter 0]

21 Feb 2023

Overall notes

We noticed error messages were appearing where the figure and table callouts were supposed to be; we’ve now fixed those instances. We’ve also fixed some values in Tables 2 (via tracked changes).

Responses to Reviewer 1 comments

Abstract

1.) Quantitative results should be mentioned in the abstract. I suggest at least to mention accuracy and precision, linearity, and recovery.

Inter-day accuracy and precision, linearity, and recovery of the four described methods were included in the abstract, as word limit permitted.

2.) In the abstract, I miss data of an application to real human samples. This should be added.

A swift description of the application of the four methods to clinical investigation was added to the abstract, as word limit permitted.

Introduction

3.) Lines 37-40. Is it really necessary to mention the awkward IUPAC name of the compound? The structure in figure 1 should be sufficient in my opinion.

The IUPAC name of lurbinectedin has been removed.

Materials and methods

4.) Line 113 and chapter 2.6. For what kind of human plasma has the method been developed and validated (K3-EDTA, or else)?

The kind of plasma (K3-EDTA) has been specified in section 2.6.

5.) Lines 89 and 231. The references [10,11] are not correct. The guidelines of EMA and FDA are missing in the references list. The others seem to be ok.

Wrong references were removed and replaced by correct ones (FDA and EMA method validation guidelines)

6.) Line 325. Again, what kind of plasma. Did you test different anticoagulations?

Results and Discussion

Only one anticoagulation (K3-EDTA) has been tested. This has been clarified in line 237.

7.) Lines 363-373. Please describe more exactly why the metabolites could not be included in the method for the parent compound.

An explanation of why the metabolites were quantified with different methods, was included.

8.) Figure 2. Please mention the concentration of the patient plasma sample. When comparing the peak areas, it seems as if the patient sample is >50 ng/mL?

The concentration (34.3 ng/mL) of the patient plasma sample which chromatogram is displayed in Fig 2, has been added at the title of Fig 2.

9.) Lines 572-581. With regard to the application of the methods to patients, please add a respective chapter in the Materials and Methods part and present at this position (chapter 3.6) only the results of the patient application.

A subsection in the Materials and Methods section has been added, and text from the Results section has been split.

Responses to Reviewer 2 comments

Specific points:

1. Certain key information is missing on lurbinectidin plasma protein binding: is the unbound fraction independent of plasma concentrations? What is the impact of hypoalbuminemia on fu? Indeed, the authors have evaluated low and high alpha1—glycoprotein acid plasma levels on fu, but abnormal albuminemias occur at least as often.

The in vitro PPB experiments to determine lurbinectedin fu described in this manuscript were conducted in three experimental conditions, varying lurbinectedin, AAG and albumin concentrations:

• Condition 1: fixed lurbinectedin, varying AAG, and fixed albumin.

• Condition 2: fixed lurbinectedin, fixed AAG, and varying albumin.

• Condition 3: varying lurbinectedin, fixed AAG, and fixed albumin.

Unbound fraction did not change in condition 2 and 3. This findings suggest that the unbound fraction is independent of lurbinectedin and albumin plasma concentrations. This is described in Section 3.5 (In vitro PPB experiments)

2. What about the plasma protein binding of the two main metabolites (M4 and M6)?

The PK of these two main metabolites of lurbinectedin are been characterized in the clinical pharmacology studies because they were also the most abundant lurbinectedin-related products in toxicokinetic studies. However, their relative exposure compared to the parent in patients is low so, in principle, their contribution to lurbinectedin activity is not deem relevant.

The plasma protein binding experiments with lurbinectedin were mostly driven to calculate the dissociation constants of lurbinectedin with AAG and albumin, in order to estimate individual unbound lurbinectedin concentration from total lurbinectedin concentration, AAG and albumin, to be used in exposure-response analyses. Since unbound exposure of metabolites is not expected to impact lurbinectedin activity, their plasma protein binding has not been studied so far. 

3. Conclusions. The clinical relevance of DDI corresponding to the plasma protein binding displacement of lurbinectidin —a drug with a rather low hepatic extraction ratio— should be discussed or at least mentioned. Lastly, reflecting the poor development of the Discussion, the list of references is too limited.

A paragraph addressing this aspect was added in the Results and Discussion section (in Section 3.6 Method application).

4. Introduction (page 4, lines 66-68): It seems that the PK/PD analysis was performed using fu estimated from plasma AAG and albumin levels to correct total lurbinectedin plasma concentrations. The main results should be given somewhere, compared with non-corrected results.

The advantage of using fu in ER models with lurbinectedin has been included in the Results and Discussion section (Methods Application).

5. The choice of some of the drugs used to evaluate the selectivity of this assay should be justified. If doxorubicin, and its metabolite, is logical considering the combination of lurbinectedin plus doxorubicin —as is the case for antiemetic drugs, e.g.—, that of platinum compounds seems unlikely. Why is itraconazole considered, but not rifampicin?

The compound used as CYP3A4 inducer in the dedicated DDI study is not rifampicin but bosentan, which has been evaluated for selectivity and is already mentioned in the manuscript.

6. Calibration curves (Page 16): What about the intercept concerning the carry-over phenomenon?

During the validation a small amount of carry-over was observed in the first sample following the ULOQ standard, to prevent any possible effect pre-treatment samples were never injected directly after Cmax samples without “zero” samples (blank matrix with IS) being added after the Cmax samples. In the general PK profiles were injected without concern for this as the amount of carryover was insignificant compared to the size of the following peak.

---

## [Decision Letter · Decision Letter 1]

16 Mar 2023

Quantitative determination of lurbinectedin, its unbound fraction and its metabolites in human plasma utilizing ultra-performance LC–MS/MS

PONE-D-22-19338R1

Dear Dr. Nichola King,

We’re pleased to inform you that your manuscript has been judged scientifically suitable for publication and will be formally accepted for publication once it meets all outstanding technical requirements.

Kind regards,

Tommaso Lomonaco, Ph.D

Academic Editor

PLOS ONE

Reviewers' comments:

Reviewer's Responses to Questions

**Comments to the Author**

1. If the authors have adequately addressed your comments raised in a previous round of review and you feel that this manuscript is now acceptable for publication, you may indicate that here to bypass the “Comments to the Author” section, enter your conflict of interest statement in the “Confidential to Editor” section, and submit your "Accept" recommendation.

Reviewer #1: All comments have been addressed

Reviewer #2: All comments have been addressed

2. Is the manuscript technically sound, and do the data support the conclusions?

Reviewer #1: Yes

Reviewer #2: Yes

3. Has the statistical analysis been performed appropriately and rigorously? 

Reviewer #1: N/A

Reviewer #2: Yes

4. Have the authors made all data underlying the findings in their manuscript fully available?

Reviewer #1: Yes

Reviewer #2: Yes

5. Is the manuscript presented in an intelligible fashion and written in standard English?

Reviewer #1: Yes

Reviewer #2: Yes

6. Review Comments to the Author

Reviewer #1: The authors addressed my comments adequately. I do not have further question, remarks, or comments.

Reviewer #2: no additional comment to revised Ms

.........................................................................

7. PLOS authors have the option to publish the peer review history of their article (what does this mean?). If published, this will include your full peer review and any attached files.

Reviewer #1: No

Reviewer #2: **Yes: **étienne Chatelut

---

## [Editor Report · Acceptance letter]

23 Mar 2023

PONE-D-22-19338R1 

Quantitative determination of lurbinectedin, its unbound fraction and its metabolites in human plasma utilizing ultra-performance LC–MS/MS 

Dear Dr. King:

I'm pleased to inform you that your manuscript has been deemed suitable for publication in PLOS ONE. Congratulations! Your manuscript is now with our production department. 

Kind regards, 

on behalf of

Dr. Tommaso Lomonaco 

Academic Editor

PLOS ONE